# Planar Boronic Graphene and Nitrogenized Graphene Heterostructure for Protein Stretch and Confinement

**DOI:** 10.3390/biom11121756

**Published:** 2021-11-24

**Authors:** Xuchang Su, Zhi He, Lijun Meng, Hong Liang, Ruhong Zhou

**Affiliations:** 1Department of Physics, Hangzhou Dianzi University, Hangzhou 310018, China; mr.su.su@outlook.com (X.S.); lianghongstefanie@163.com (H.L.); 2Institute of Quantitative Biology, College of Life Sciences, Zhejiang University, Hangzhou 310027, China; hezhi32@aliyun.com (Z.H.); ljmeng@zju.edu.cn (L.M.); 3Department of Chemistry, Columbia University, New York, NY 10027, USA

**Keywords:** molecular dynamics simulation, boronic graphene, nitrogenized graphene, inplane heterostructure, protein stretch and confinement

## Abstract

Single-molecule techniques such as electron tunneling and atomic force microscopy have attracted growing interests in protein sequencing. For these methods, it is critical to refine and stabilize the protein sample to a “suitable mode” before applying a high-fidelity measurement. Here, we show that a planar heterostructure comprising boronic graphene (BC_3_) and nitrogenized graphene (C_3_N) sandwiched stripe (BC_3_/C_3_N/BC_3_) is capable of the effective stretching and confinement of three types of intrinsically disordered proteins (IDPs), including amyloid-β (1–42), polyglutamine (Q42), and α-Synuclein (61–95). Our molecular dynamics simulations demonstrate that the protein molecules interact more strongly with the C_3_N stripe than the BC_3_ one, which leads to their capture, elongation, and confinement along the center C_3_N stripe of the heterostructure. The conformational fluctuations of IDPs are substantially reduced after being stretched. This design may serve as a platform for single-molecule protein analysis with reduced thermal noise.

## 1. Introduction

Protein sequencing at the single-molecule level is crucial for personalized medicines and the detection of post-translational modifications in proteins [1,2,3,4]. Recently, several single-molecule techniques such as atomic force microscopy (AFM) [5,6], quantum tunneling [7,8,9], and nanopore [3,10,11] have been proposed for protein sequencing, which allow the direct read-out of structural differences of individual amino acids. Although promised to be with high accuracy and low cost, a large gap still resides between these proof-of-principle methods and the ultimate sensitivity for the discrimination of 20 different amino acids. One major challenge is the noisy signals caused by the thermal fluctuations of amino acids [12,13]. Moreover, proteins usually possess coiled or folded conformations in a solution, which imposes difficulties in the analysis of the atomic structure of protein [13,14]. Therefore, to put the single-molecule protein sequencers into potential commercial use, the controllable manipulation (such as elongation) and confinement of the protein conformation are prerequisites.

Current nanochannel and nanopore sequencing techniques naturally provide steric confinement for analytes, and the single-molecular sensitivity can be realized with the cross-section of confinement in the same order of magnitude as the size of the amino acids [15,16]. However, the narrow cross-section inevitably causes a large entropy barrier that hampers protein capture into the nanoscale channel [15,16]. The non-specific interaction between the protein and the nanostructure can also affect the precision of measurement and induce clogging [4,12]. To address these issues, a planar two-dimensional (2D) heterostructure has recently been proposed for biomolecular capture, stretching, and confinement [17,18,19]. The planar 2D heterostructure can be fabricated by the seamless stitching of two 2D materials (for example, graphene and hexagonal boron nitride) with a similar lattice constant [20,21,22,23]. As the key mechanism for this heterostructure to manipulate protein conformation is the adsorption energy contrast for a protein molecule on different 2D materials [18,24], the performance should depend on the type of 2D material selected.

Boronic graphene (BC_3_) and nitrogenized graphene (C_3_N) are two new types of graphene derivatives that have been successfully synthesized [25,26]. Both BC_3_ and C_3_N exhibit excellent structural stability and share very similar honeycomb lattices [24,25], making them suitable to form planar heterojunctions. On the other hand, with differently doped heteroatoms (boron and nitrogen), BC_3_ and C_3_N have demonstrated a distinct contrast of binding affinities for biomolecules [27], which can be harnessed for biomolecular manipulation. Owing to these features, we are highly motivated to design a BC_3_/C_3_N/BC_3_ in-plane heterostructure (Figure 1) for protein stretching and confinement. To study the interaction mechanism between the heterostructure and protein, three representative intrinsically disordered proteins (IDPs), including amyloid-β (Aβ_1–42_), polyglutamine (polyQ_42_), and α-synuclein (α-Syn_61–95_) are taken as examples. Utilizing all-atom molecular dynamics (MD) simulations, we show that the disordered conformations of IDPs can be stretched into a linear manner along the C_3_N stripe sandwiched between two BC_3_ domains. This highly regular and confined conformation might be suitable for analysis by single-molecule methods such as AFM [5,6] and quantum tunneling [7,8,9]. Moreover, the conformational fluctuations of proteins can be significantly reduced after being stretched and energetically confined on the C_3_N stripe. The periodic atomic charge distributions on BC_3_ also induce the formation of high-density water clusters on the BC_3_ surfaces, which may further provide steric hindrances to restrict the conformational fluctuation of IDPs. The insights from our study might benefit the improvement of the signal-to-noise ratio for single-molecule protein analysis.

## 2. Method

We used molecular dynamics (MD) simulations to simulate a two-dimensional (2D) sandwich BC_3_/C_3_N/BC_3_ planar heterostructure with a total size of 16.2 × 14.1 nm^2^ [2] (Figure 1). Among them, the width of the C_3_N stripe seamlessly spliced between the two BC_3_ sheets is 1.2 nm. The force fields of BC_3_ and C_3_N can be obtained by referring to the previous research [28], in which the lattice constants of both are 2.5 Å, and the boron and nitrogen atoms have partial charges of 0.378 e and −0.168 e, respectively. To maintain the overall and local charge neutrality of the planar heterostructure, the carbon atoms in BC_3_ and C_3_N carry −0.126 e and 0.056 e, respectively [29].

Following the similar approach in our previous studies [17,24,30,31,32], we performed a pre-equilibrium simulation of the conformation of each IDP fragment from solution to adsorption on a BC_3_ nanosheet, and then extended the BC_3_ nanosheet (along with the peptide) to construct the BC_3_/C_3_N/BC_3_ planar heterostructure. After that, the entire system is placed in a box with a size of 16.2 × 14.1 × 5.0 nm^3^ [3] and solvated with 100 mM KCl electrolyte, which contains approximately 66,000 atoms. In addition, to explore the difference in hydrophilic or hydrophobic properties between BC_3_ and C_3_N, we constructed two systems of BC_3_ or C_3_N in a water box with a size of 4.9 × 4.2 × 5.0 nm^3^ [3]. Later, to characterize the interface behavior of water molecules on the planar heterostructure, we additionally constructed a heterostructure-water system with a box size of 16.2 × 14.1 × 5.0 nm^3^ [3].

The Gromacs software package [33] (version 5.1.4) was used for our MD simulations, and VMD [34] was used for trajectory visualization. The TIP3P model [35] was used for water molecules, the CHARMM36 force field [36] for proteins/peptides, and standard force fields for ions. Following the scheme used in many previous, similar researches [30,37,38,39,40,41], we used the LINCS algorithm to constrain the covalent bonds with hydrogen atoms, with a time step of 2 fs. The particle mesh Ewald (PME) method [42] with a grid size of about 1 Å was used to calculate the long-range electrostatic interactions, while the smooth cut-off method was used for the van der Waals (vdW) interactions, with a cut-off distance of 1.2 nm. Periodic boundary conditions were used in all three-dimensional directions. The Parrinello-Rahman algorithm [43] was applied in the z-direction with a semi-isotropic pressure coupling of 1 bar, and the V-rescale thermostat [44] was used to control the simulation temperature at 300 K. Then, under the NPT ensemble, several independent 400 ns trajectories were generated for each system for data collection. In all simulations, except that the atoms in the two-dimensional planar heterostructure are frozen, all other atoms can move freely.

## 3. Result

Some representative snapshots (Figure 2) were shown for the trajectories of Aβ, polyQ, and α-Syn on the BC_3_/C_3_N/BC_3_ planar heterojunction, respectively. Taking Aβ (Figure 2A) as an example, starting from the curled and folded structure adsorbed on the BC_3_ domain, Aβ first diffused to the C_3_N region quickly, arriving at t = ~22 ns, which is consistent with our previous findings on the similar free and rapid diffusions of adsorbents on other two-dimensional material plane [24]. Finally, at t = ~250 ns, Aβ was stretched on the C_3_N stripe and maintains a straightened conformation. Following the same procedure, we also checked the trajectories of polyQ (Figure 2B) and α-Syn (Figure 2C) and found that although they had slightly different dynamic behaviors in the initial diffusion and subsequent stretching phases, they all displayed the same straightened conformation as Aβ on the C_3_N stripe at the end. In addition, we analyzed the average end-to-end distance normalized by the residue number (*L/N*) for each protein (Figure 2D). During the simulation, the *L/N* of Aβ, polyQ, and α-Syn increased from ~1.0 Å to 2.2 Å, 2.2 Å, and 2.3 Å, respectively. It is worth noting that although the *L/N* values of these IDPs fluctuate sharply due to their high flexibility before being fully stretched, they maintain their respective highest values during the last 100 ns simulations, indicating that IDPs can be spontaneously stretched on the C_3_N stripe. It is also worth noting that the fluctuations of *L/N* are reduced after the peptides are stretched (Figure 2D). To characterize the change of conformational fluctuations of peptides in the stretching process, we further conducted MD simulations of peptides in free solution, on the BC_3_ surface, and the BC_3_/C_3_N/BC_3_ surface, respectively. Figure 3 illustrates the probability distributions of *L/N* of Aβ peptides in different environments. The distribution of *L/N* of Aβ confined in the heterostructure has a much narrower width than those in other environments. Moreover, distributions of *L/N* of polyQ (Appendix A) and a-Syn (Appendix A) also demonstrate similar results. The above analyses indicate that the conformational fluctuations of proteins can be significantly reduced when they are confined on the C_3_N stripe BC_3_/C_3_N/BC_3_.

To reveal the stretching mechanism of IDPs on the BC_3_/C_3_N/BC_3_ planar heterostructure, we adopted Aβ as an example to examine the details of the interaction between peptides and BC_3_/C_3_N/BC_3_ (Figure 4). During the simulation, the average number of contact atoms per Aβ residue with BC_3_ decreased from ~9 to ~4, while the average number of contact atoms per Aβ residue with C_3_N increased from 0 to ~8 (Figure 4A). Here, we defined a contact when any atom in BC_3_/C_3_N/BC_3_ was within 4.0 Å of any heavy atom of the protein. As shown in the scatter plot of Figure 4B, in the stretching process, the peptide contacted more with C_3_N while less with BC_3_, accompanied by a gradual increase in the magnitude of interaction energy. According to the above results, the peptide interacted more strongly with C_3_N than BC_3_, which drove its stretching on BC_3_/C_3_N/BC_3_. To further understand the physical mechanism of the driven process, we scanned the interaction energy Δ*E* between the peptide and BC_3_/C_3_N/BC_3_ by moving the stretched conformation of Aβ horizontally and rigidly from the C_3_N stripe to the BC_3_ domain. As shown in Figure 4C, when the peptide was on the C_3_N stripe, the van der Waals interaction (vdW) energy presented a narrow energy well with a depth of −1.6 kcal per mol per residue (black dotted line). It is also worth noting that the average value of Coulomb interaction energy between Aβ and the BC_3_/C_3_N/BC_3_ was almost zero (red line) due to the local charge neutrality of the planar heterostructure. Moreover, we repeated all of the above analyses for the polyQ (Appendix A) and α-Syn (Appendix A) on BC_3_/C_3_N/BC_3_. Similarly, the straightening and restriction of polyQ and α-Syn on the narrow C_3_N stripe were also mainly due to the differences in the vdW interactions between the polypeptides and C_3_N and BC_3_.

To further investigate the influence of the C_3_N stripe width on the efficiency of protein stretching, we also constructed BC_3_/C_3_N/BC_3_ heterostructures with the stripe widths ranging from 0.6 nm to 1.8 nm and scanned the vdW interaction energy between the elongated Aβ peptide and each heterostructure, respectively (Appendix A). Appendix A shows the potential wells of C_3_N stripes with different widths, and Appendix A shows the “potential-depth” and the “potential-width” at a half-minimum of each well as a function of the C_3_N stripe width. For both curves of the depths and widths of potential wells, the inflection points occur when the C_3_N layer width is equal to 1.2 nm. After this deflection point, the increasing stripe width would lead to a slower decrease of the depth of the potential well, and a faster increase of the width of the potential well. It is noteworthy that a potential well with a deeper depth and narrower width can lead to a better performance of the protein stretching. Considering the optimization of both the depth and the width of the potential well, the C_3_N layer width of 1.2 nm is recommended for efficient protein stretching.

Moreover, in practice, the interface between BC_3_ and C_3_N may not be perfect, as shown in Figure 1, and rough domains with mixed units of BC_3_ and C_3_N might exist at the interface. One simple way to illustrate this effect is to use the 1.2-nm-width C_3_N stripe in our current simulation as the base and use the width of the 1.0-nm- to 1.4-nm-C_3_N stripes as the upper and lower bounds, as the adsorption potential well of the stripe with rough interfaces should be in-between the wells of these two widths. As shown above in Appendix A, the potential wells of the 1.0-nm-width, 1.2-nm-width, and 1.4-nm-width C_3_N stripes share similar shapes, with potential-well depths of −1.4 kcal/mol, −1.6 kcal/mol, and −1.7 kcal/mol respectively. Thus, the roughness of the interface is not likely to affect much on the potential well for protein stretching and confinement.

In addition, it is known that the behavior of water on the surface of 2D materials also has an important influence on the interaction between protein and 2D materials in the aqueous environment. Therefore, we further analyzed the behaviors of interfacial water on BC_3_ and C_3_N respectively, and the influence of these interfacial waters on the confinement of IDPs on the planar heterostructure (Figure 5). From the water density maps (Figure 5A,B) along the normal (*Z*) directions of 2D materials, the first water solvation shells of BC_3_ and C_3_N were both located on Z = ±0.35 nm. More interestingly, we found a periodic enhancement of water density clusters between the BC_3_ layer and its first solvation shell, which were not observed clearly on the C_3_N layer. The presence of these water clusters on BC_3_ might be attributed to the strong but nonuniform partial charge distributions (+0.378 e for a boron atom and −0.126 e for a carbon atom). While on C_3_N, the partial charges (−0.168 e for a carbon atom and +0.056 e for a nitrogen atom) were much smaller, and not strong enough to induce noticeable water clusters between the 2D layer and the first solvation shell. Meanwhile, we further calculated the two-dimensional water density map along the surface (*XY* direction) of BC_3_/C_3_N/BC_3_ (only water molecules within 0.25 nm in the *Z* direction of the 2D plane were counted). As shown in Figure 5C,D, these water clusters were periodically distributed on the BC_3_ domain, which could form a steric hindrance (Figure 5E) that hindered the diffusion of IDPs’ residues to the BC_3_ domain. These interfacial water molecules might further strengthen the restriction on the linear conformation of IDPs.

## 4. Conclusions

In this work, we applied molecular dynamics (MD) simulation to study the stretching process of several representative IDPs on the 2D sandwiched BC_3_/C_3_N/BC_3_ planar heterostructure. The IDPs could be spontaneously straightened and then restricted along the C_3_N stripes. Moreover, we have shown that the conformational fluctuations of IDPs were significantly reduced when IDPs were confined on the C_3_N stripe. The protein stretching and confinement were mainly driven by the stronger adsorption potential of C_3_N than that of BC_3_. Additionally, we found that the interfacial water molecules on the BC_3_ surface might further act as steric hindrances to enhance the restriction of protein. This linearly confined structure on the 2D surface may be feasible for scanning tunneling microscope (STM) [45,46] and atomic force microscope (AFM) [5,6] to identify the amino acids in the protein. Furthermore, compared with other “hard” nano-confinements such as nanochannels and nanopores, this heterostructure provides “soft” confinement that is based on the adsorption potential difference between the two 2D nanomaterials. Considering the large entropy barrier for stretching a coiled or folded protein to a linear conformation, this soft, energetic confinement allows some protein residues to temporarily move outside the center C_3_N stripe while “regulating” the protein conformation, thus smoothly overcoming the entropy barrier in the stretching process. Therefore, this heterostructure holds the potential to be coupled with nanopore- and nanochannel-sensing methods to ease the clogging problem. The delivery of the stretched protein samples from this heterostructure to a nanochannel can be driven by a pressure-driven flow, for example. Last but not least, C_3_N and BC_3_ have been reported to possess higher biocompatibility than graphene [28]. Our work may also offer insight into the design of biocompatible nanodevices.

## Figures and Tables

**Figure 1 biomolecules-11-01756-f001:**
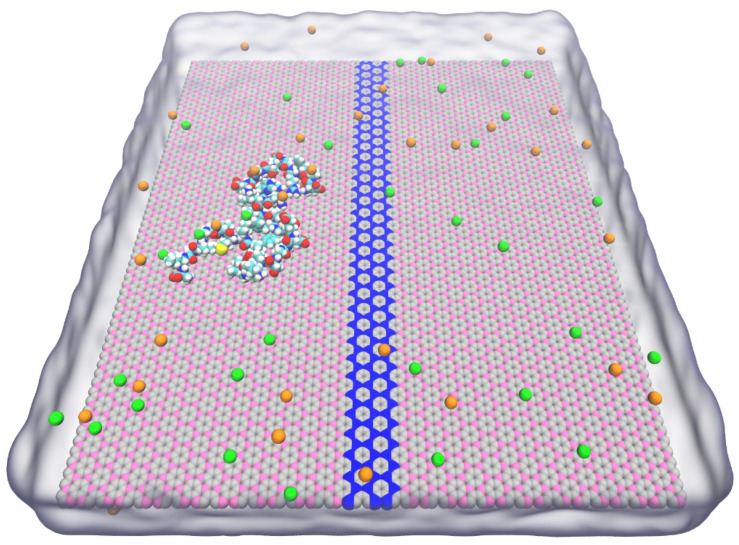
The initial simulation configuration of the Aβ_1–42_ peptide on the BC_3_/C_3_N/BC_3_ heterostructure. Carbon, boron, and nitrogen atoms in the 2D material are colored in silver, pink, and blue, respectively. Atoms in the peptide are shown as spheres (C: cyan; O: red; H: white; N: blue; and S: yellow). K^+^ and Cl^−^ ions are colored in green and orange, while water molecules are shown as glass bubbles.

**Figure 2 biomolecules-11-01756-f002:**
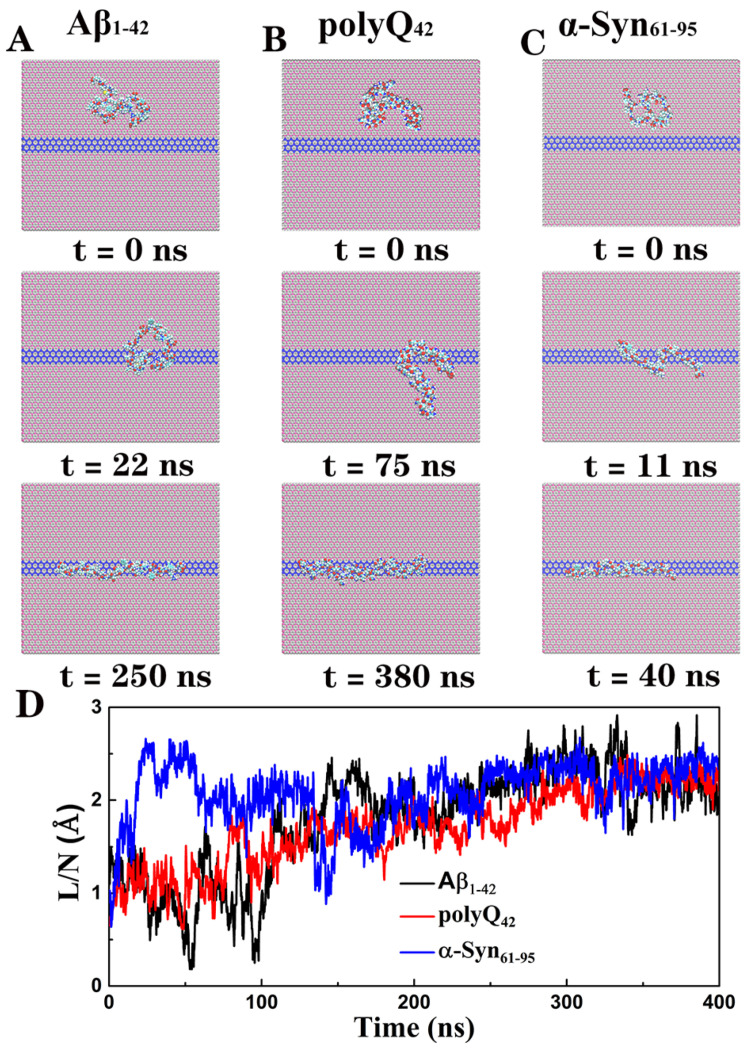
Representative snapshots of stretching processes of (**A**) Aβ_1–42_, (**B**) polyQ_42_, and (**C**) α-Syn_61–95_ on BC_3_/C_3_N/BC_3_. (**D**) End-to-end distances for Aβ (black line), polyQ (red line), α-Syn (blue line) during the simulations.

**Figure 3 biomolecules-11-01756-f003:**
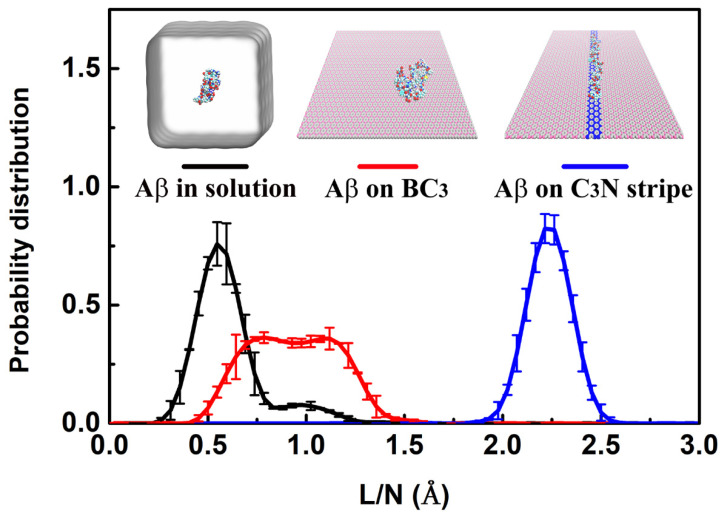
Probability distribution of end-to-end distances for Aβ in solution (black line), on BC_3_ surface (red line), and on C_3_N stripe (blue line).

**Figure 4 biomolecules-11-01756-f004:**
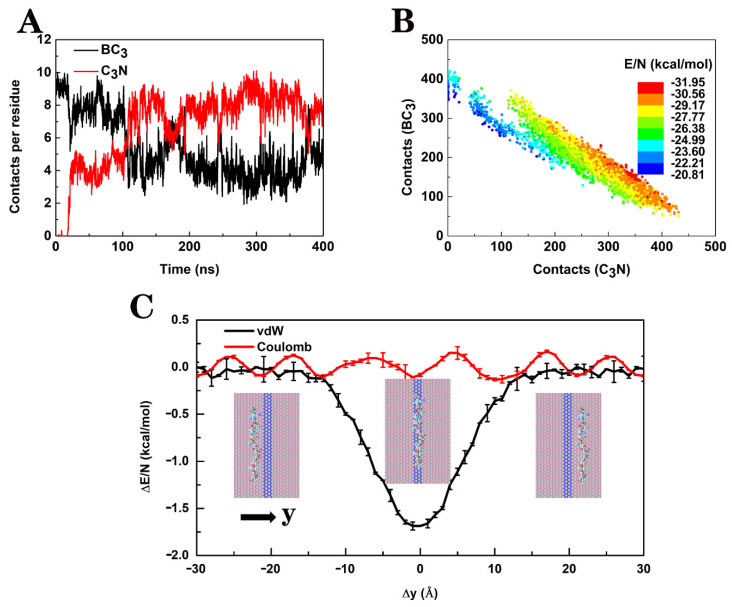
(**A**) Number of atoms in BC_3_/C_3_N/BC_3_ that are within 4.0 Å of the Aβ peptide throughout the simulation. (**B**) A scatter plot of the number of atoms in BC_3_/C_3_N/BC_3_ in contact with Aβ. The color represents the average interaction energy per residue between Aβ and BC_3_/C_3_N/BC_3_. (**C**) The average van der Waals (black line) and Coulomb (red line) interaction energy between Aβ and BC_3_/C_3_N/BC_3_ with standard deviations, when the elongated Aβ peptide moved across the C_3_N band in the y-direction.

**Figure 5 biomolecules-11-01756-f005:**
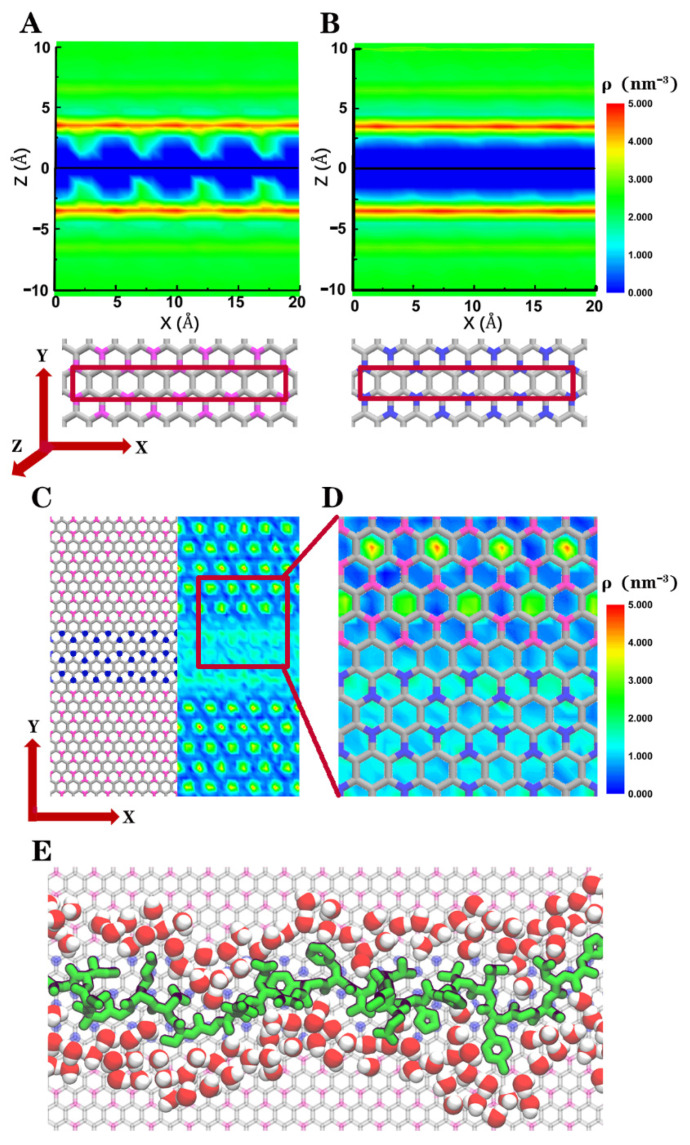
The density maps of water along the normal directions of the (**A**) BC_3_ and (**B**) C_3_N planes, where the 2D materials are located at Z = 0. As shown in the structure below, the calculation was performed along the zigzag direction, while only water molecules inside the red square were calculated. (**C**) Two-dimensional water density map on the BC_3_/C_3_N/BC_3_ surface; water molecules within ±0.25 nm in the *Z* direction were counted. (**D**) Magnified map of water density at the boundary of BC_3_ and C_3_N. (**E**) Straightened Aβ peptide on BC_3_/C_3_N/BC_3_ with interfacial water surrounded; only water molecules within 0.5 nm of both the heterostructure and protein are shown.

## Data Availability

Not applicable.

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
