# Peer review of "Planar Boronic Graphene and Nitrogenized Graphene Heterostructure for Protein Stretch and Confinement"

_biomolecules, 2021, doi:10.3390/biom11121756_

Round 1
Reviewer 1 Report
The article “Planar Boronic Graphene and Nitrogenized Graphene Heterostructure for Protein Stretch and Confinement” is aimed at molecular dynamics simulations of stretching several representative intrinsically disordered proteins on 2D sandwiched BC3/C3N/BC3 planar heterostructure. In general, the paper can be regarded as good; it is definitely able to attract target readers to make them keep an eye on this side of science. Modeling different structures, applicable in biocompatible nanodevices, can help the synthetics all over the world to find new perspective directions in this field of knowledge. I believe this work deserves due attention, and also I would like to recommend it to publish in Biomolecules journal. However, I would kindly ask the authors to give answers to some of the following remarks and questions.
The way I see the issue, the authors proposed a perfect case where the interface between the BC3 and C3N layers is smooth and clear. It seems to me that reaching such a high quality of the interface could be almost impossible as a practical matter. There follows a question: how crucial is the presence of ‘flaws’ on the BC3 and C3N interface be for efficient protein stretching? Here, ‘flaw’ refers to the case when BC3 and C3N units are swapped; therefore, the interface represents a domain of a certain width between the layers where BC3 and C3N units are randomly mixed.
Also, due to the perfection of the modeled case, it is unclear how the width change of the C3N layer being enclosed between the BC3 layers can affect the efficiency of protein stretching, e.g. if the C3N layer is rough, with its width varying from 1 to 1.4 nm in different areas.
Why did you choose the C3N layer width equal to 1.2 nm? How efficient would the protein stretching be if the width layer value was changed either up or down?
If there were the answers to these questions in the article, then you would be able to forecast performance criteria of nanodevices based on the proposed two-dimensional multilayered BC3/C3N/BC3 planar heterostructure and also hypothesize the fundamental possibility of its creation.
Reviewer 2 Report
The present manuscript entitled “Planar Boronic Graphene and Nitrogenized Graphene Hetero-structure for Protein Stretch and Confinement” authored by Xuchang Su et al. describe the planar heterostructure comprising boronic graphene (BC3) and nitrogenized graphene (C3N) sandwiched strips (BC3/C3N/BC3) is capable for effective stretching and confinement of three types of intrinsically disordered proteins including amyloid-β (1–42), polyglutamine (Q42) and α-Synuclein (61–95). Furthermore, the molecular dynamics simulations exhibit that the protein molecules interact more strongly with the C3N strip than the BC3 one, which leads to its capture, elongation, and confinement along the center C3N stripe of the heterostructure. The result analysis is very accurate and adequate, lacks of major errors. Therefore, I would recommend the publication of the manuscript in the “Biomolecules” after some Minor improvements.
I advise the authors to consider the following points while revising their manuscript.
Comment 1: Abstract is poorly written, revise it properly.
Comment 2: Some relevant references in this area are still missing in the introduction section, so include some significant relevant references from recent years. The novelty of the work must be highlighted, in the last paragraph of the introduction.
Comment 3: This manuscript was poorly written, a lot of English grammar mistakes were observed. The authors must be careful with the use of English. The manuscript needs to be checked again to avoid typographical and grammatical mistakes. Apart from this, several typos in the text were observed. Each thing must be carefully checked and improved. which can be further polished by the native English speakers.
Comment 4: Properly revise the conclusion as per the findings.
Reviewer 3 Report
Manuscript ID: Biomolecules-1462148
Comments and Suggestions for Authors
The introduced manuscript under the title of “Planar Boronic Graphene and Nitrogenized Graphene Heterostructure for Protein Stretch and Confinement” introduces the interaction mechanism between the heterostructure and protein and shows that the disordered conformations of IDPs that can be stretched into a linear manner along the C3N stripe sandwiched between two BC3 domains. This highly regular and confined conformation might help future single-molecule protein sequencing with high signal-to-noise ratio for personalized medicines and detection of post-translational modifications in proteins.
Publication of this paper may be suitable because the topic is interesting, the manuscript is well organized, the results are interesting. However, the references should be updated, and number of references seems too much. These modifications should be included before publication.
Following suggestions to the authors may be proposed to improve the paper:
- The references should be updated or supported by more recent publications.
- The number of references should be reduced.
Round 2
Reviewer 1 Report
Thanks for the answers to the questions. I hope they helped improve the article.